# THINK WHEN YOU NEED: SELF-ADAPTIVE CHAIN-OF-THOUGHT LEARNING

## ABSTRACT

Chain of Thought (CoT) reasoning enhances language models' performance but often leads to inefficient "overthinking" on simple problems. We identify that existing approaches directly penalizing reasoning length suffer from hyperparameter sensitivity and limited generalizability, especially for fuzzy tasks where ground truth is unavailable. Our approach constructs rewards through length and quality comparisons, guided by theoretical assumptions that jointly enhance solution correctness with conciseness. Our methodology extends naturally to both verifiable tasks with definitive answers and fuzzy tasks requiring subjective evaluation. Experiments across multiple reasoning benchmarks demonstrate that our method maintains accuracy while generating significantly more concise explanations, effectively teaching models to "think when needed."

## 1 INTRODUCTION

As reasoning models, particularly OpenAI o1 (Jaech et al., 2024) and Deepseek-R1 (Guo et al., 2025), gain widespread adoption, the use of Chain of Thought (CoT) reasoning has garnered increasing attention. Numerous studies (Jaech et al., 2024; Guo et al., 2025; Team et al., 2025) have demonstrated that incorporating CoT in reinforcement learning (RL) is crucial for improving reasoning performance. Moreover, CoT exhibits strong generalization capabilities, suggesting that robust CoT abilities in one domain can enhance performance in other areas requiring reasoning skills. However, a significant drawback when using reasoning models is their inefficiency with simple questions—they often consume excessive time generating elaborate explanations even when simpler answers would suffice. Existing works (Team et al., 2025; Arora & Zanette; Aggarwal & Welleck, 2025; Luo et al., 2025a) mainly focus on adding penalty on length itself, which calls for careful design and hyperparameter setting when incorporating with other rewards. Actually, some questions are straightforward and require brief responses, while others are complex and necessitate detailed explanations. Hence, uniformly penalizing all long responses can compromise performance on more challenging problems. Furthermore, existing algorithms lack clear explanation and theoretical support.

In this paper, we propose a novel reward algorithm to address the aforementioned limitations. Our approach diverges from conventional methods by calculating rewards based on pairwise relationships between samples rather than explicitly penalizing response length. Specifically, we first establish comprehensive pairwise reward assumptions and corresponding rules applicable across various scenarios. We then systematically compare all possible sample combinations and compute pairwise rewards for each comparison. The final reward for each sample is determined by aggregating all pairwise rewards involving that particular sample. This relational definition of rewards enables natural integration with other pairwise reward mechanisms (e.g., true-false comparative rewards), supported by clear theoretical assumptions and explanatory conditions. Furthermore, our investigation encompasses both verifiable tasks with definitive ground truth and fuzzy tasks where objective truth is unavailable. Through extensive experiments across diverse settings, we rigorously validate the effectiveness of our proposed methodology according to well-defined evaluation criteria, demonstrating significant improvements over existing approaches.

In summary, our contributions in this work include:

- Introducing a novel efficient learning algorithm that maintains performance while reducing reasoning length. Our algorithm design is guided by theoretical assumptions, making it easily generalizable and compatible with other reward structures.
- Extending the algorithm to fuzzy tasks, offering the first efficient learning approach for scenarios where ground truth is unavailable, addressing both pairwise and pointwise reward settings.
- Conducting extensive experiments across diverse reasoning benchmarks and fuzzy task setting to validate the effectiveness and generalizability of our proposed algorithm.

## 2 METHODOLOGY

In this section, we introduce a general method that accommodates customized comparative scenarios. We first address the verifiable task setting, establishing key theoretical assumptions and developing corresponding pairwise reward rules. We then extend our algorithms to fuzzy tasks, presenting approaches for both pairwise and pointwise reward scenarios.

### 2.1 PROPOSED METHOD

Our proposed method leverages pairwise comparisons to establish a robust reward framework for reinforcement learning. Given a set of $L$ possible comparison scenarios and $N$ sampled responses, we define rewards through systematic pairwise evaluations rather than absolute scoring.

For any two samples $m_i$ and $m_j$ ($0 \leq i, j < N$) that satisfy a specific pairwise scenario $s_l$ ($0 \leq l < L$), we assign rewards according to the following formulation:

$$(r_{ij}(m_i), r_{ij}(m_j)) = (\gamma_l^+, \gamma_l^-) \quad \text{if} \quad (m_i, m_j) \in s_l,$$

where $m_i$ represents the positive sample receiving reward $\gamma_l^+$, and $m_j$ represents the negative sample receiving reward $\gamma_l^-$, both determined by the specific scenario $s_l$. Each scenario $s_l$ represents a distinct comparative relationship between samples (e.g., one response being correct while another is incorrect).

The total reward for each sample $m_i$ is then computed by aggregating all pairwise rewards it receives when compared with every other sample in the set:

$$r(m_i) = \sum_{k \neq i} r_{ik}(m_i). \tag{1}$$

Here, $r_{ik}(m_i)$ denotes the reward assigned to sample $m_i$ when compared with sample $m_k$. The condition $k \neq i$ ensures that self-comparisons are excluded from the calculation. This approach allows us to determine relative reward values across the entire sample space, incorporating multiple comparison dimensions simultaneously.

### 2.2 VERIFIABLE TASK SETTING

We evaluate the efficient learning approach on reasoning tasks where ground truth answers are available and verifiable. In this context, responses are classified as either correct or incorrect.

**Assumption 1.** *Correct answers receive higher rewards than incorrect answers. All incorrect answers receive identical rewards.*

Based on this assumption, we establish the following pairwise scenarios $s_0$ and $s_1$:

- Pairwise scenario $s_0$: Both responses are incorrect. Both responses receive a reward of 0, i.e., $\gamma_0^+ = \gamma_0^- = 0$.
- Pairwise scenario $s_1$: One response is correct and the other is incorrect. The correct response receives a reward of $\alpha$ ($\alpha > 0$) while the incorrect one receives a reward of $-\alpha$, i.e., $\gamma_1^+ = \alpha, \gamma_1^- = -\alpha$.

Next, we characterize the relationship between response length and reward as follows:

**Assumption 2.** *Among correct answers, shorter responses receive higher rewards than longer ones. Correct answers of equal length receive equal rewards.*

Based on this assumption, we define the pairwise scenarios $s_2$ and $s_3$:

- Pairwise scenario $s_2$: Both responses are correct but have different lengths. The shorter response receives a reward of $\beta$ ($\beta > 0$) and the longer one receives $-\beta$, i.e., $\gamma_2^+ = \beta, \gamma_2^- = -\beta$.
- Pairwise scenario $s_3$: Both responses are correct and have identical lengths. Both responses receive a reward of 0, i.e., $\gamma_3^+ = \gamma_3^- = 0$.

Note that in our framework, $\gamma^+ + \gamma^- = 0$ for simplicity, though this constraint can be relaxed for more complex scenarios. The algorithm details are illustrated in Figure 1.

To calculate the cumulative reward for each sample using Equation 1, we simplify by ignoring scenario $s_3$ (assuming all correct responses have different lengths) and setting $\beta = 1$. Among $N$ samples, suppose there are $M$ ($0 \leq M \leq N$) incorrect samples. From comparisons with correct samples:

- Each incorrect response receives a reward of $-\alpha \times (N - M)$.

For correct samples, when ranked by length from longest to shortest, the rewards are calculated as:

- The longest correct response reward: $(1 + \alpha)M - N + 1$.
- The second longest correct response reward: $(1 + \alpha)M - N + 3$.
- Each subsequent shorter correct sample gains 2 additional reward points compared to the previous one.
- The shortest correct response reward: $(\alpha - 1)M + N - 1$.

We make the following assumption that the penalty for incorrect responses should exceed the reward differential between correct answers of varying lengths.

**Assumption 3.** *The penalty for incorrect responses should be more severe than the reward difference between long and short correct responses.*

To satisfy this assumption, the following inequality must hold:

$$(\alpha - 1)M + N - 1 - ((1 + \alpha)M - N + 1) < ((1 + \alpha)M - N + 1) - (-\alpha(N - M)).$$

Simplifying both sides:

$$3M + (\alpha - 3)N + 3 > 0 \quad \forall N, M.$$

Given that $0 \leq M \leq N$, this inequality is satisfied when:

$$\alpha > \frac{3N - 3}{N} = 3 - \frac{3}{N},$$

when $M = 0$.

Finally, we ensure that even after penalization for length, correct responses still receive higher rewards than incorrect ones.

**Assumption 4.** *The lowest reward achieved by any correct response exceeds the highest reward achieved by any incorrect response.*

In a set of $N$ samples, the minimum reward a correct response can receive is $\alpha - (N - 2)$, occurring when there is only one incorrect sample (providing reward $\alpha$) and all other correct samples are shorter (each imposing a penalty of $-1$). Conversely, the maximum reward an incorrect response can receive is $-\alpha$, occurring when there is only one correct sample. Note that we exclude the cases where all responses are correct or all are incorrect. For Assumption 4 to hold, we need

$$\alpha - (N - 2) > -\alpha.$$

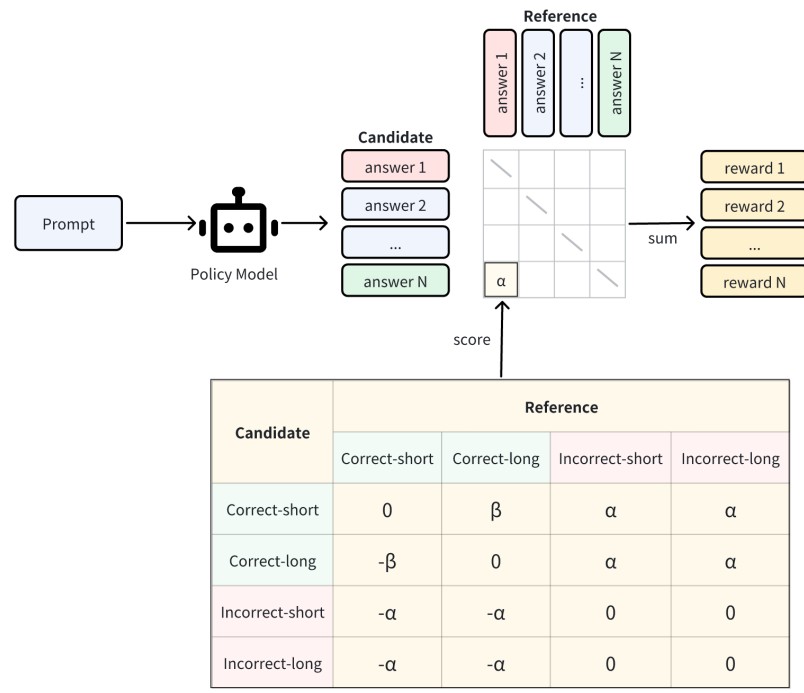

Figure 1: Illustration of the proposed algorithm and verifiable task comparison. The approach involves sampling $N$ candidate answers and performing comprehensive pairwise comparisons between them. The final reward for each candidate is computed as the summation of all pairwise rewards obtained when compared against the other answers.

| Candidate | Reference | | | |
| --- | --- | --- | --- | --- |
| | Better-short | Better-long | Worse-short | Worse-long |
| Better-short | \ | \ | \ | α+β |
| Better-long | \ | \ | α | \ |
| Worse-short | \ | -α | \ | \ |
| Worse-long | -α-β | \ | \ | \ |

Figure 2: Fuzzy task comparison scenarios. When compare short better and long worse answers, short better answer obtain reward of $\alpha + \beta$ while the long worse one receives the $-\alpha - \beta$. When compare long better and short worse answers, long better answer obtain reward of $\alpha$ while the short worse one receives the $-\alpha$.

Simplifying above inequality and we obtain

$$\alpha > \frac{N-2}{2}.$$

This constraint ensures the hierarchical integrity of our reward system across all possible sample configurations with mixed correctness. In practice, we introduce artificial positive and negative samples to address the corner case that all responses are either uniformly correct or incorrect to maintain assumption validity.

### 2.3 FUZZY TASK SETTING

We further extend our approach on fuzzy tasks where ground truth answers are unavailable.

#### 2.3.1 PAIRWISE REWARD

In this setting, each response receive comparative signals by direct comparison with other responses.

**Assumption 5.** *Better answers receive higher rewards than worse answers.*

Based on this assumption, we establish the following pairwise scenario $f_0$:

- Pairwise scenario $f_0$: The better response receives a reward of $\alpha$ ($\alpha > 0$) while the worse response receives $-\alpha$.

Next, we characterize the relationship between response length and reward:

**Assumption 6.** *Longer worse responses incur additional penalties compared to shorter better ones.*

Based on this assumption, we specify the pairwise scenario $f_1$:

- Pairwise scenario $f_1$: If the better response is shorter, it receives $\alpha + \beta$ while the worse response receives $-\alpha - \beta$.

Since we have two comparison metrics (response quality and response length), we make the following assumption to establish comparison priority:

**Assumption 7.** *Assumption 5 still holds after applying the length penalty from Assumption 6, i.e., after all comparisons, better answers still receive higher rewards than worse answers after accounting for length penalties.*

The extreme case occurs when, among $N$ responses, the best response is the longest, receiving a reward of $(N-1)\alpha$, while the second-best response is the shortest, receiving reward of $(N-2)(\alpha + \beta) - \alpha$. To satisfy Assumption 7, we expect the best response still receives higher reward:

$$(N-1)\alpha > (N-2)(\alpha + \beta) - \alpha.$$

Simplifying this inequality yields

$$\beta < \frac{2\alpha}{N-2}.$$

If we set $\beta = 1$, then we obtain

$$\alpha > \frac{N-2}{2},$$

which is the same inequality as we obtain in Assumption 4.

#### 2.3.2 POINTWISE REWARD

In this setting, each response $i$ receives an independent score $s_i$ from a Bradley-Terry Reward Model (BTRM). The final reward $r(i)$ for each response $i$ is defined as:

$$r(i) = s_i - \frac{c(i)}{N} \cdot d(i),$$

where $N$ is the group size, $d(i)$ is the minimum score difference between this response and any lower-scored response, and $c(i)$ represents the number of responses that have shorter length but equal or better scores. The algorithm detail is available in Appendix A.

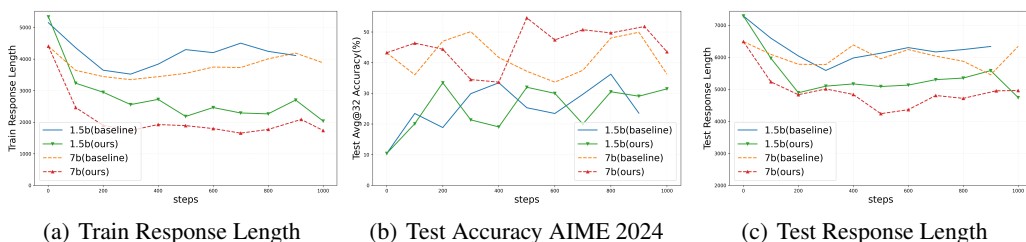

(a) Train Response Length     (b) Test Accuracy AIME 2024     (c) Test Response Length

Figure 3: Performance comparison in the DeepScaleR setting. Our method achieves comparable test accuracy to the baseline while significantly reducing response length during both training and testing phases. Solid lines represent results for the 1.5B model, while dashed lines represent the 7B model.

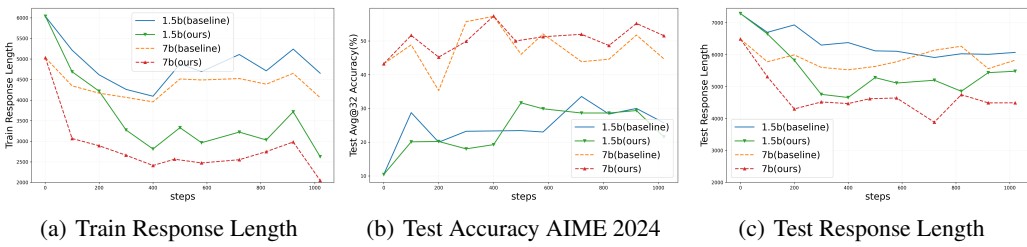

(a) Train Response Length     (b) Test Accuracy AIME 2024     (c) Test Response Length

Figure 4: Performance comparison in the 8K DAPO setting. Our method maintains or slightly improves test accuracy compared to the baseline while substantially reducing response length. Solid lines represent results for the 1.5B model, while dashed lines represent the 7B model.

## 3 EXPERIMENTAL RESULTS

In this section, we present a comprehensive evaluation of our proposed method. We first describe our experimental setup and implementation details, followed by detailed analyses of results across various configurations for verifiable tasks.

### 3.1 VERIFIABLE TASK

We implement our approach using Group Relative Policy Optimization (GRPO) (Shao et al., 2024) as the reinforcement learning algorithm with a group size $N = 8$, leveraging the open-source VeRL framework (Sheng et al., 2024). Based on our theoretical formulation, we set the correctness reward parameter $\alpha = 5$ and the length reward parameter $\beta = 1$. We evaluate our method on DeepSeek-R1-Distill-Qwen-1.5B and DeepSeek-R1-Distill-Qwen-7B models (Yang et al., 2024; Guo et al., 2025) with 8K/16K maximum response lengths. All experiments utilize the AdamW optimizer with a learning rate of $1 \times 10^{-6}$, a prompt batch size of 128, and a mini-batch size of 64 during rollout for parameter updates.

In accordance with Assumption 4, we incorporate one artificial incorrect sample and one artificial correct sample with maximum length during training. These samples serve exclusively for pairwise reward calculation within batches and are excluded when computing advantage.

For our evaluation pipeline, we utilize training datasets from DeepScaleR (Luo et al., 2025b) and DAPO (Yu et al., 2025), and assess performance across five benchmarks. To ensure robust evaluation metrics, we evaluate each test instance 32 times and report avg@32 results, with all evaluation inferences using a temperature setting of 0.6. We omit training reward results since our pairwise reward formulation is symmetric, naturally centering rewards around zero. All experiments were conducted on a computational cluster comprising 4 machines with 8 GPUs each, running for approximately 72 hours to complete 1000 training steps.

### 3.1.1 DEEPSCALER SETTING

Figure 3 illustrates our algorithm's performance on both DeepSeek-R1-Distill-Qwen-1.5B and DeepSeek-R1-Distill-Qwen-7B models using the DeepScaleR-Preview-Dataset (Luo et al., 2025b)

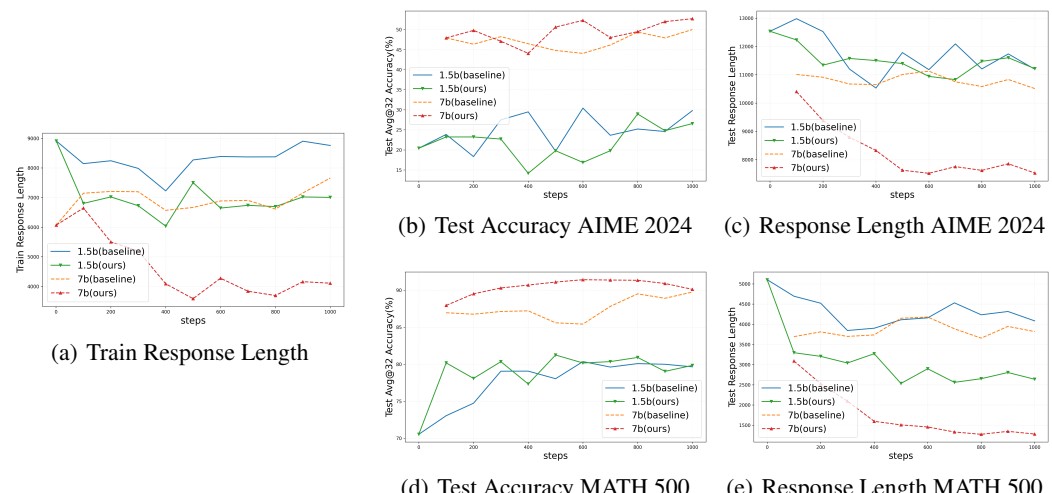

(b) Test Accuracy AIME 2024    (c) Response Length AIME 2024

(a) Train Response Length

(d) Test Accuracy MATH 500    (e) Response Length MATH 500

Figure 5: Performance comparison in the 16K DAPO setting across different benchmarks. Our method maintains comparable test accuracy while substantially reducing response length across both AIME 2024 and MATH 500 benchmarks. Solid lines represent the 1.5B model, while dashed lines represent the 7B model.

(MIT license). Results for the 7B model are represented by dashed lines, while the 1.5B model results are shown with solid lines.

As training progresses, our algorithm achieves substantial reduction in response lengths, decreasing from over 4,000 to under 2,000 tokens, while the baseline maintains lengths around 2,500 tokens. Critically, test accuracy remains comparable between our method and the baseline, despite generating significantly shorter responses during evaluation. This pattern is consistent across both model sizes, demonstrating that our method effectively reduces response length while preserving solution quality.

### 3.1.2 DAPO SETTING

We further evaluate our approach using the DAPO-MATH-17K dataset (Yu et al., 2025) on both model architectures. Figure 4 presents results with an 8K token maximum response length constraint. For the 7B model, our approach reduces training response length to approximately 2,500 tokens compared to over 4,000 tokens for the baseline. Test accuracy remains comparable between methods, with our approach even achieving slightly better performance in later training stages. For test response length, our method demonstrates consistent reduction, decreasing from approximately 6,000 to 4,500 tokens. Similar improvements are observed with the 1.5B model. As shown in Table 1 for the 1.5B model, our algorithm reduces average response length by 30% while improving test accuracy on 3 out of 5 benchmarks.

When extending the maximum response length to 16K tokens (Figure 5), our algorithm dramatically reduces the average response length from approximately 11K to 8K tokens for the 7B model while maintaining comparable test performance across both model sizes. Notably, the 7B model exhibits greater response length reduction than the 1.5B model, suggesting that larger models with enhanced capabilities can more effectively compress reasoning steps without sacrificing performance.

A comparison of test results between AIME 2024 and MATH 500 reveals an adaptive behavior in our algorithm: it enables the 1.5B model to reduce response length on the relatively easier MATH 500 benchmark, while maintaining longer responses for the more challenging AIME 2024 problems. This demonstrates that our approach effectively implements the principle of "think when needed," adaptively adjusting reasoning depth based on problem complexity. Additional evaluation metrics are provided in Appendix B.1.

| Evaluation | Average Response Length | | | Test Accuracy (%) | |
|---|---|---|---|---|---|
| | Baseline | Ours | $\Delta$ (%) | Baseline | Ours |
| AIME 2024 | 6031 | 4653 | $-22.85$ | **28.0** | **28.0** |
| AMC | 4594 | 3358 | $-26.90$ | **65.0** | 63.0 |
| MATH 500 | 2567 | 1480 | $-42.35$ | 82.5 | **85.0** |
| Minerva | 3136 | 1581 | $-49.59$ | 26.4 | **27.4** |
| Olympiad Bench | 4360 | 3323 | $-23.78$ | 45.3 | **45.6** |
| Average | 4137 | 2879 | $-30.41$ | 49.4 | **49.8** |

Table 1: Comparison of algorithms for response length and test accuracy with 8K maximum length for 1.5B model. $\Delta$ indicates the percentage reduction in response length by our method compared to baseline.

## 3.2 FUZZY TASK

In this section, we examine our approach on fuzzy tasks—scenarios where ground truth solutions are unavailable, necessitating alternative evaluation strategies. We utilize the AlpacaFarm dataset (Dubois et al., 2023) (Apache-2.0 license) for training and evaluation. For these experiments, we employ the GRPO algorithm with a group size $N = 4$. For each prompt, we generate 4 candidate responses and leverage the AlpacaEval framework (Li et al., 2023) to compute pairwise rewards between response combinations. To enhance computational efficiency, we substitute the original GPT-4 with DeepSeek-R1 as our generative reward model. Consistent with our verifiable task experiments, we evaluate performance on DeepSeek-R1-Distill-Qwen-1.5B and DeepSeek-R1-Distill-Qwen-7B models. We maintain the same optimization parameters: AdamW optimizer with learning rate $1 \times 10^{-6}$, a prompt batch size of 32, and mini-batch size of 32 for updates.

For evaluation methodology, given the absence of ground truth answers, we assess performance through relative comparisons between models using our reward model. We quantify **relative advantage** as *(win-loss)/total*, where a value of 0 indicates equivalent performance between compared models, and +1 signifies our model's complete dominance across all test samples. We compare our trained models against both the baseline model at equivalent training steps and the original SFT model to demonstrate the efficacy of the reinforcement learning approach. All experiments enforce a maximum response length constraint of 1024 tokens.

### 3.2.1 ALPACAFARM RESULTS

Following the theoretical framework established in Assumption 7, we implement two experimental configurations. In our primary configuration, we set preference reward $\alpha = 5$ and length reward $\beta = 3$. Results are presented in Figure 6. The data reveal that our method's advantage over the SFT model increases progressively with training steps, demonstrating effective optimization. Simultaneously, when compared to the baseline at equivalent training steps, our approach maintains comparable performance for the 1.5B model and achieves approximately a 10% relative advantage for the 7B model. Notably, our method dramatically reduces the chain-of-thought (CoT) length to near-zero for both model sizes, compared to the baseline's sustained average length of approximately 1000 characters for the 1.5B model and 1500 characters for the 7B model.

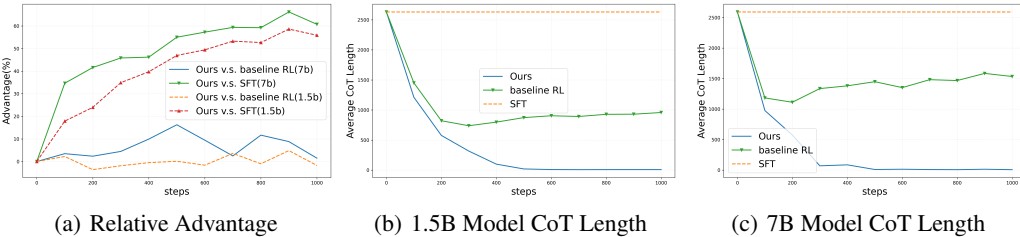

(a) Relative Advantage     (b) 1.5B Model CoT Length     (c) 7B Model CoT Length

Figure 6: Performance analysis on AlpacaFarm with reduced length penalty $\alpha = 5, \beta = 3$.

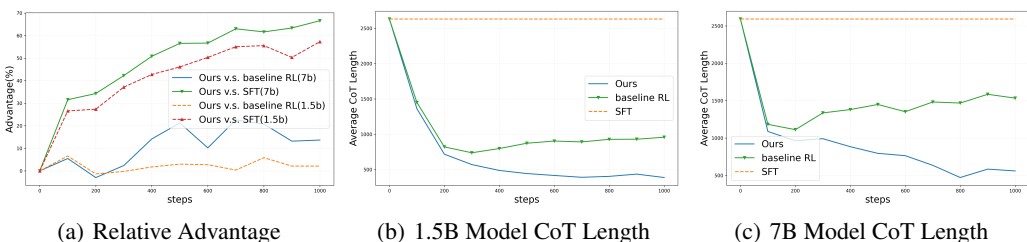

(a) Relative Advantage     (b) 1.5B Model CoT Length     (c) 7B Model CoT Length

Figure 7: Performance analysis on AlpacaFarm with reduced length penalty $\alpha = 5, \beta = 1$.

We further investigate the impact of varying the length penalty by implementing a second configuration with $\alpha = 5$ and reduced length reward $\beta = 1$. As illustrated in Figure 7, this configuration results in a more gradual reduction in CoT length compared to the higher $\beta$ setting, but yields more pronounced relative advantages over both baseline and SFT models. These findings suggest that higher $\beta$ values facilitating more aggressive length reduction when response conciseness is a priority.

## 4 RELATED WORKS

### 4.1 EFFICIENT CoT METHODS

Existing works primarily focus on controlling response length directly for efficiency. Aggarwal & Welleck (2025) introduced Length Controlled Policy Optimization (LCPO), a reinforcement learning approach that simultaneously optimizes for both accuracy and adherence to user-specified length constraints. Similarly, Luo et al. (2025a) proposed Length-Harmonizing Fine-Tuning (O1-Pruner), which aims to minimize reasoning overhead while preserving model accuracy. Their method establishes the model's baseline performance through pre-sampling before applying RL-based fine-tuning to encourage more concise reasoning processes under accuracy constraints. Alternative approaches include the "Long2short" algorithm by Team et al. (2025), which applies penalties to response length regardless of answer correctness. Chen Li (2025) refined this approach by selectively applying length penalties only to incorrect responses, preserving the natural length of successful reasoning paths. A notable limitation of these methods is their explicit dependence on raw length measurements, necessitating careful hyperparameter tuning when combining with other reward functions. Furthermore, these approaches often lack robust theoretical foundations explaining why and how length constraints affect model performance and reasoning capabilities.

### 4.2 PAIRWISE REWARD MODELS

Traditional reinforcement learning typically assigns rewards to individual samples using models like the Bradley-Terry reward model (BTRM). However, BTRM requires training an additional model and necessitates calibration. In the pairwise reward domain, Jiang et al. (2023) introduced PAIRRANKE, which employs a specialized pairwise comparison method to distinguish subtle differences between candidate outputs. More recently, pairwise reward models (Liu et al., 2025) have been developed specifically for Best-of-N sampling scenarios. In knockout tournament settings, PairJudge reward models conduct pairwise judgments between candidate solutions and iteratively eliminate incorrect ones, improving the efficiency of identifying optimal responses.

## 5 CONCLUSION

In this work, we introduce a general pairwise reward framework that successfully addresses Chain of Thought inefficiency in language models by structuring rewards through relative comparisons rather than absolute metrics. Our approach significantly improves reasoning efficiency without sacrificing accuracy, as demonstrated through extensive experiments across various reasoning benchmarks and fuzzy tasks. The theoretical analysis of our method enable straightforward adaptation to diverse tasks, offering valuable insights for future research in efficient language model reasoning.

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

# Supplementary Materials

## A  POINTWISE REWARD

Given point rewards for $N$ samples, we denote the score assigned by BTRM to the $i$-th response as $s_i$ for $i \in \{1, 2, \ldots, N\}$. Let $l_i$ represent the length of the $i$-th response.

For any sample $i$ with score $s_i$ and length $l_i$, we first define the minimum score gap between this sample and any lower-scored sample:

$$d(i) = \min_{j \neq i, s_i > s_j} (s_i - s_j)$$

To penalize unnecessarily verbose responses, we identify samples that achieve equal or higher scores with shorter lengths. For each sample $i$, we define:

$$c(i) = |\{j : s_j \geq s_i \text{ and } l_j < l_i\}|$$

where $|\cdot|$ denotes the cardinality of the set.

The modified reward for each sample is then computed as:

$$r(i) = s_i - \frac{c(i)}{N} \cdot d(i)$$

This formulation ensures that the penalty applied to verbose responses is proportional to both the score gap $d(i)$ and the number of more efficient responses $c(i)$, while remaining bounded by the smallest relevant score difference.

For the corner case where sample $k$ has the lowest score (i.e., $k = \arg\min_i(s_i)$), $d(k)$ is not defined by the original formula. In this case, we define it as the average minimum score gap across all other samples:

$$d(k) = \text{avg}_{j \neq k} d(j)$$

## B  EXPERIMENTAL RESULTS

### B.1  DIFFERENT BENCHMARKS

In this part, we present detailed evaluation results for our approach across various mathematical reasoning tasks. We report both testing accuracy and response length metrics to demonstrate the effectiveness of our method.

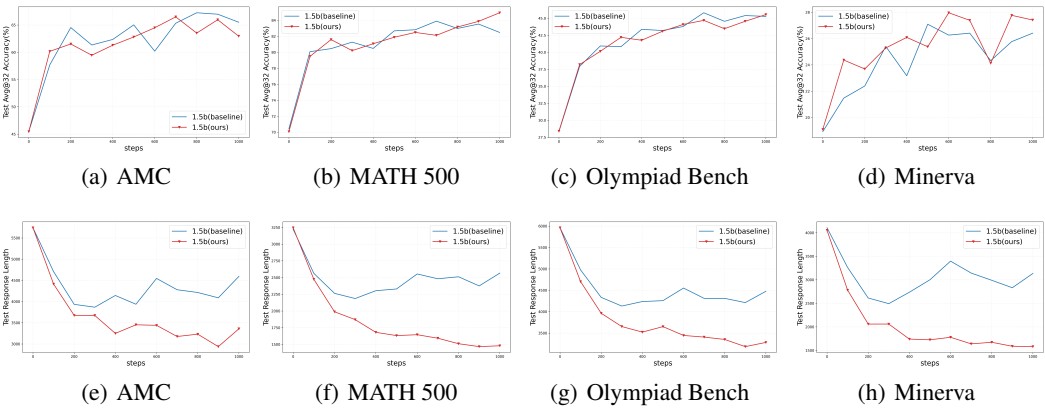

Figure 8: 1.5B Model Results on DAPO setting with 8K maximum response lengths.

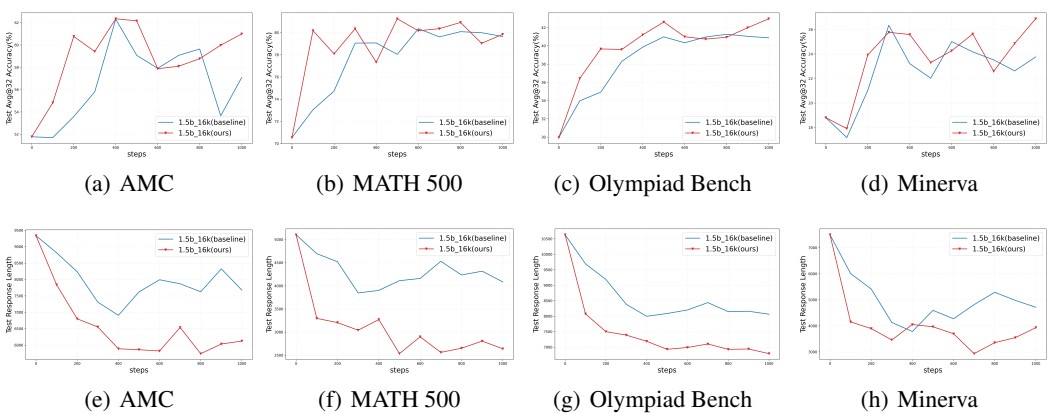

Figure 9: 1.5B Model Results on DAPO setting with 16K maximum response lengths.

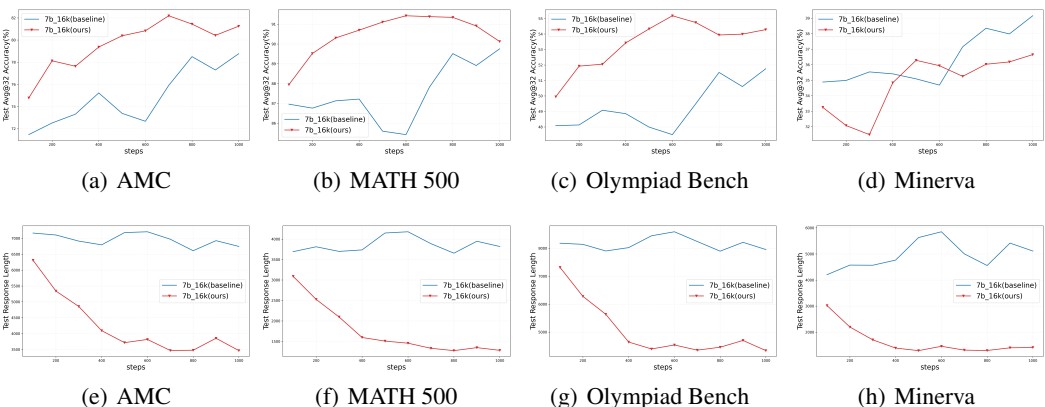

Figure 10: 7B Model Results on DAPO setting with 16K maximum response lengths.

Figure 8 demonstrates that our algorithm achieves testing accuracy comparable to baseline methods for the 1.5B model with 8K maximum response length, while significantly reducing response lengths across all benchmark datasets. Similar patterns are observed in Figure 9, where the maximum length is extended to 16K tokens, further validating our approach's effectiveness at maintaining performance while producing more concise solutions.

The results for the 7B model, shown in Figure 10, exhibit even more pronounced length reduction while maintaining comparable accuracy. This supports our hypothesis that larger models can achieve superior performance with more concise reasoning, and that our algorithm provides greater efficiency benefits as model size increases. The consistent performance across different model sizes and context length configurations demonstrates the robustness of our approach in optimizing both solution accuracy and conciseness.

## B.2 DIFFERENT HYPERPARAMETERS

We test the algorithm with different hyperparameters in this part. The accuracy for the verifiable task is averaged over five benchmarks. Reported results use the 1.5B model; for the verifiable task we run 400 steps, while for the fuzzy task we run 200 steps.

From Table 2, for the verifiable task the theoretical requirement derived in Section 2 implies $\alpha > 3$ when $\beta = 1$ (here $N = 8$). Settings that do not satisfy this condition yield lower accuracy than the baseline. We also observe that CoT length decreases when increasing $\beta$ or reducing $\alpha$.

Table 2: Verifiable task: Accuracy (%) and CoT length (tokens) under different $(\alpha, \beta)$ (1.5B).

| Metric | Baseline | 10/1 | 5/1 | 3/1 | 1/1 | 5/10 |
|---|---|---|---|---|---|---|
| Accuracy (%) | 47.9 | 48.6 | 48.3 | 45.1 | 43.0 | 36.6 |
| CoT length (tokens) | 4589.0 | 3322.0 | 3226.0 | 2335.0 | 1822.0 | 1522.0 |

**Note:** Column shorthand $a/b$ stands for $\alpha = a, \beta = b$.

Table 3: Fuzzy task: Win rate (%) and CoT length (tokens) under different $(\alpha, \beta)$ (1.5B).

| Metric | 5/1 | 5/3 | 3/1 | 1/1 | 10/1 |
|---|---|---|---|---|---|
| Win rate vs. baseline (%) | 48.3 | 47.1 | 50.7 | 52.1 | 51.1 |
| CoT (tokens) | 718.0 | 577.0 | 701.0 | 438.0 | 770.0 |
| Baseline (tokens) | 800.0 | 800.0 | 800.0 | 800.0 | 800.0 |

**Note:** Column shorthand $a/b$ stands for $\alpha = a, \beta = b$.

For the fuzzy task (Table 3), the theoretical requirement is $\alpha > 1$ when $\beta = 1$ (here $N = 4$). The win rate remains approximately $50\%$ across different reward parameters, indicating the performance does not degrade substantially with parameter changes. Moreover, CoT length decreases as $\alpha$ decreases, which shows that our fuzzy-task parameterization is robust to varying $\alpha$.

In conclusion, $\alpha$ should be set to satisfy the theoretical requirement. Moreover, small $\alpha$ is usually preferred since it can reduce CoT length without harming performance.

### B.3 COMPARISON WITH OTHER BASELINES

We further compare our method (TWYN) with another length-reduction algorithm, Long2Short Team et al. (2025), on the verifiable task.

Table 4: 1.5B model: Accuracy (%) and CoT length (tokens).

| Algorithm | Baseline | TWYN (ours) | Long2Short Team et al. (2025) |
|---|---|---|---|
| Accuracy (%) | 47.9 | 48.3 | 47.7 |
| CoT Length (tokens) | 4589.0 | 3226.0 | 2946.0 |

**Note:** Best trade-off is judged by both accuracy and CoT reduction.

Table 5: Algorithm comparison on Math-500 (reported).

| 1.5B | Base Model | Accuracy (%) | CoT (tokens) |
|---|---|---|---|
| 1.5B Baseline | DeepSeek-R1-Distill-Qwen-1.5B | 83.9 | 2492.0 |
| THINKPRUNE Hou et al. (2025) | DeepSeek-R1-Distill-Qwen-1.5B | 83.9 | 2209.0 |
| **TWYN (ours)** | DeepSeek-R1-Distill-Qwen-1.5B | 85.0 | 1480.0 |
| **7B** | Base Model | Accuracy (%) | CoT (tokens) |
| 7B Baseline | DeepSeek-R1-Distill-Qwen-7B | 89.7 | 3764.0 |
| DAST Shen et al. (2025) | DeepSeek-R1-Distill-Qwen-7B | 93.6 | 3309.0 |
| TWYN (ours) | DeepSeek-R1-Distill-Qwen-7B | 91.6 | 1479.0 |

**Remark:** CoT lengths are measured in tokens.

**Reported results on Math-500.** Compared to Long2Short, Long2Short achieves slightly shorter CoT length but degrades accuracy relative to baseline because it imposes length penalties even on incorrect responses. TWYN improves accuracy relative to the baseline while substantially reducing

CoT length. Against THINKPRUNE, TWYN yields higher accuracy and shorter CoTs. Compared to DAST on the 7B model, DAST attains higher accuracy, but TWYN produces much shorter CoT lengths (roughly less than half).

### B.4 DIFFERENT ARCHITECTURES

We evaluate TWYN on different architectures and larger models. Results are shown below.

Table 6: LLaMA-8B: Baseline vs. TWYN.

| Metric | Baseline | TWYN |
|---|---|---|
| Accuracy (%) | 56.04 | 56.79 |
| CoT Length (tokens) | 4631 | 4022 |

Table 7: Qwen-32B: TWYN vs. Baseline.

| Metric | TWYN v.s. Baseline |
|---|---|
| Win Rate (%) | 54.70 |
| CoT Length (tokens) | 1132 v.s. 1323 |

These results indicate that across different architectures and larger models, our algorithm continues to improve performance while reducing CoT length.

### B.5 RESULTS ON TASKS THAT REQUIRE DEEP THINKING

We evaluated the baseline and TWYN on the GPQA-diamond benchmark, which is highly challenging and requires deep multi-step reasoning with 32k inferences on 1.5B model.

Table 8: GPQA-diamond: Baseline vs. TWYN ($\alpha = 5, \beta = 1$).

| Algorithm | Baseline | TWYN ($\alpha = 5, \beta = 1$) |
|---|---|---|
| Accuracy (%) | 26.8 | 27.3 |
| CoT Length (tokens) | 4489 | 2908 |

The result shows that TWYN outperforms the baseline on this difficult dataset while producing substantially shorter CoT. Note that our length penalty is applied only to overly long *correct* responses; incorrect responses are not penalized for brevity. This design prevents the model from exploiting a trivial strategy of producing short but incorrect outputs, and instead encourages correct and efficient reasoning.

## C CASE STUDY

We present a case study to illustrate how the thought process changes. Below we show the prompt and the model's chain-of-thought (CoT) for both the Baseline and TWYN.

PROMPT.

Suppose $f(x)$ is an invertible function, and suppose that $f(2) = f^{-1}(2) = 4$.

What is the value of $f(f(2))$? Let's think step by step. Write the thought in <think> and </think>, then output the final answer within $\Box$.

BASELINE (COT).

<think>Okay, so I have this problem here: Suppose $\\( f(x) \\)$ is an
invertible function, and it's given that $\\( f(2) = f^{-1}(2) = 4$
$\\)$. I need to find the value of $\\( f(f(2)) \\)$. Hmm, let's break
this down step by step.\n\nFirst, I know that $\\( f \\)$ is
invertible, which means that $\\( f \\)$ is a one-to-one function.
That also implies that $\\( f \\)$ has an inverse function $\\( f^{-1}$
$\\)$, and by definition, $\\( f(f^{-1}(x)) = x \\)$ and $\\($
$f^{-1}(f(x)) = x \\)$ for all $\\( x \\)$ in the appropriate
domains.\n\nNow, let's parse the given information: $\\( f(2) = 4 \\)$
and $\\( f^{-1}(2) = 4 \\)$. Hmm, that's interesting. So, $\\( f \\)$
evaluated at 2 gives 4, and the inverse function evaluated at 2 also
gives 4. \n\n**Wait a minute, let's recall what $\\( f^{-1}(x) \\)$
means**. The inverse function $\\( f^{-1}(x) \\)$ essentially reverses
the action of $\\( f(x) \\)$. So, if $\\( f(a) = b \\)$, then $\\($
$f^{-1}(b) = a \\)$. Is that correct? Let me make sure I have this
straight. Yes, that's right. So, if $\\( f \\)$ maps 2 to 4, then $\\($
$f^{-1} \\)$ should map 4 back to 2. \n\nBut in the problem, it's
given that $\\( f^{-1}(2) = 4 \\)$. That seems a bit counterintuitive
at first because if $\\( f(2) = 4 \\)$, then $\\( f^{-1}(4) = 2 \\)$,
not $\\( f^{-1}(2) = 4 \\)$. So, there must be some connection here
that I'm missing.\n\nLet me write down what I know:\n\n1. $\\( f(2) =$
$4 \\)$\n2. $\\( f^{-1}(2) = 4 \\)$\n\nFrom the first point, $\\( f(2) =$
$4 \\)$ means that when I input 2 into the function $\\( f \\)$, I get 4
as the output. \n\nFrom the second point, $\\( f^{-1}(2) = 4 \\)$
means that when I input 2 into the inverse function $\\( f^{-1} \\)$,
I get 4 as the output. But as I recalled earlier, the inverse
function essentially reverses the input and output of the original
function. So, if $\\( f(a) = b \\)$, then $\\( f^{-1}(b) = a \\)$.
\n\nApplying that to the second point: $\\( f^{-1}(2) = 4 \\)$ should
imply that $\\( f(4) = 2 \\)$. Because if $\\( f^{-1}(2) = 4 \\)$, then
$\\( f(4) = 2 \\)$. That makes sense now. So, $\\( f(4) = 2$
$\\)$.\n\n**Wait, so let me summarize:**\n\n- $\\( f(2) = 4 \\)$
(given)\n- $\\( f^{-1}(2) = 4 \\)$ implies $\\( f(4) = 2 \\)$ (by the
definition of inverse functions)\n\nSo, now I know two specific
values of the function $\\( f \\)$: $\\( f(2) = 4 \\)$ and $\\( f(4) = 2$
$\\)$.\n\nThe problem is asking for $\\( f(f(2)) \\)$.
Let's compute that step by step.\n\nFirst, we know $\\( f(2) = 4 \\)$. So,
$\\( f(f(2)) = f(4) \\)$. But we already know from the second point
that $\\( f(4) = 2 \\)$. Therefore, $\\( f(f(2)) = f(4) = 2$
$\\)$.\n\n**Wait, let me make sure I'm not making a mistake here.**
So, $\\( f(f(2)) \\)$ is the same as $\\( f(4) \\)$ because $\\( f(2) = 4$
$\\)$. And since $\\( f(4) = 2 \\)$, then $\\( f(f(2)) = 2 \\)$.\n\nIs
there anything else I need to consider? The function is invertible,
which means it's one-to-one and onto, but I don't think that affects
this particular calculation. The key information given is $\\( f(2) =$
$4 \\)$ and $\\( f^{-1}(2) = 4 \\)$, which leads to $\\( f(4) = 2 \\)$.
Therefore, substituting back, $\\( f(f(2)) = f(4) = 2 \\)$.\n\nSo, the
value of $\\( f(f(2)) \\)$ is 2.\n\n**Final Answer**\nThe value of $\\($
$f(f(2)) \\)$ is $\\boxed{2}$.\n</think>
\n\nGiven that $\\( f(x) \\)$ is an invertible function, we know the
following:\n\n1. $\\( f(2) = 4 \\)$\n2. $\\( f^{-1}(2) = 4 \\)$\n\nFrom
the second point, since $\\( f^{-1}(2) = 4 \\)$, it implies that $\\($
$f(4) = 2 \\)$ by the definition of inverse functions.\n\nWe need to
find the value of $\\( f(f(2)) \\)$. \n\nFirst, we know $\\( f(2) = 4$
$\\)$. Therefore, $\\( f(f(2)) = f(4) \\)$.\n\nFrom the second point, we
already determined that $\\( f(4) = 2 \\)$.\n\nThus, $\\( f(f(2)) =$
$f(4) = 2 \\)$.\n\nThe value of $\\( f(f(2)) \\)$ is $\\(\\boxed{2}\\)$.

TWYN (COT).

<think>Okay, so I have this problem here: Suppose $\\( f(x) \\)$ is an
invertible function, and it's given that $\\( f(2) = f^{-1}(2) = 4$

```
\\). I need to find the value of \\( f(f(2)) \\). Hmm, let's break
this down step by step.\n\nFirst, I know that \\( f \\) is
invertible, which means that \\( f \\) is a one-to-one function.
That also implies that \\( f \\) has an inverse function, denoted as
\\( f^{-1} \\). One of the key properties of inverse functions is
that they undo each other. So, if \\( f(a) = b \\), then \\(
f^{-1}(b) = a \\). \n\nThe problem states that \\( f(2) = 4 \\). So,
applying the function \\( f \\) to 2 gives me 4. Now, it also says
that \\( f^{-1}(2) = 4 \\). Using the property of inverse functions,
if \\( f^{-1}(2) = 4 \\), then \\( f(4) = 2 \\). That's because
applying \\( f \\) to both sides of \\( f^{-1}(2) = 4 \\) gives \\(
f(f^{-1}(2)) = f(4) \\), and since \\( f \\) and \\( f^{-1} \\) are
inverses, \\( f(f^{-1}(2)) = 2 \\). Therefore, \\( f(4) = 2
\\).\n\nNow, I need to find \\( f(f(2)) \\). I already know that \\(
f(2) = 4 \\), so substituting that in, I get \\( f(f(2)) = f(4) \\).
From the earlier step, I found out that \\( f(4) = 2 \\). Therefore,
\\( f(f(2)) = 2 \\).\n\nSo, the value of \\( f(f(2)) \\) is
2.\n</think>
\n\nThe value of \\( f(f(2)) \\) is \\boxed{2}.
```

As shown above, the baseline exhibits a verbose, exploratory CoT with repeated checks and hesitations, whereas TWYN generates a concise and direct CoT that arrives at the correct answer efficiently.

