# OpenReview forum: "Think When You Need: Self-Adaptive Chain-of-Thought Learning"
_ICLR.cc/2026/Conference — Submitted to ICLR 2026_

### Official Review · Reviewer_SvRR · 2025-10-25

**Soundness:** 3
**Presentation:** 2
**Contribution:** 2
**Rating:** 2
**Confidence:** 4

**Summary:**

This paper proposes a novel and general reinforcement learning framework to addresses the critical issue of “overthinking” in LLMs, where LLMs generate unnecessarily long CoT reasoning even for simple problems. The core innovation is a pairwise reward mechanism which avoids explicit length penalties. Instead, rewards are computing by comparing samples in a group where correct answers are favored over incorrect ones, and among correct answers, shorter answers are favored over longer ones. The framwork is applied to both verifiable tasks and fuzzy tasks. The experiments on multiple mathematical reasoning benchmarks and the Alpacafarm dataset shows that the method significantly reduces reasoning length, without sacrificing accuracy.

**Strengths:**

The reward framework proposed in this paper is innovative and equally applicable to both verifiable tasks and fuzzy tasks. The method proposed in this paper is grounded in rigorous theoretical foundations. The experiments conducted on multiple diverse datasets demonstrate the effectiveness of the method, which reduces reasoning length without sacrificing accuracy.

**Weaknesses:**

1. This paper does not discuss the differences between this method and the baseline of other Long2Short methods (e.g. https://arxiv.org/pdf/2502.04463, https://arxiv.org/pdf/2502.03373, https://arxiv.org/pdf/2501.12570), only vanilla-sft and ablations of their method.
2. The experimental design and analysis on fuzzy tasks are insufficient, as detailed in the Questions section.
3. Section 2.3.2 does not introduce the relationship with the method proposed in this paper.
4. The typesetting of this paper needs adjustment, such as Figure 1 and Figure 2. It is suggested to move the extensive formula derivation section on page 3 to Appendix. There is a typo in line 809.
5. it seems like this method is relevant to previous work, which needs to be further discussed, as detailed in the Questions section.

**Questions:**

1. Could the authors compare the method proposed in this paper with the baselines of other Long2Short methods?(e.g. https://arxiv.org/pdf/2502.04463, https://arxiv.org/pdf/2502.03373, https://arxiv.org/pdf/2501.12570)
2. How is “self-adaptive” embodied in the title of this paper? Is it a characteristic of the method or the model training characteristic?
3. The evaluation results are not provided in fuzzy task. The “advantage” metric refers to a dynamic metric in reinforcement learning, rather than a performance metric of downstream tasks. Could authors clarify whether this dataset requires “thinking”? It is recommended to add more open-domain evaluation datasets that require thinking.
4. What is the underlying principle of the method? (e.g. Why does the final CoT length become near-zero in the experiment on the alpacafarm dataset?)
5.Could authors analyze why the accuracy improves when the CoT length shortens. Could authors conduct some assessments or case studies with quantifiable metrics? (e.g. https://arxiv.org/pdf/2506.12860)

---

### Official Review · Reviewer_NB13 · 2025-10-29

**Soundness:** 3
**Presentation:** 3
**Contribution:** 2
**Rating:** 4
**Confidence:** 2

**Summary:**

This paper presents a new reward strategy for reinforcement learning (RL) in language models aimed at improving Chain-of-Thought (CoT) reasoning efficiency. Instead of directly penalizing reasoning length, the method introduces a theoretically-motivated pairwise reward aggregation based on both correctness and response conciseness. Experiments across several reasoning benchmarks show that the method consistently reduces response lengths while maintaining or even improving accuracy.

**Strengths:**

1. The paper provides a clear, mathematically precise formulation of its pairwise reward mechanism, specifying assumptions and constraints with explicit derivations.
2. The method is shown to generalize across settings with and without verifiable ground truth.

**Weaknesses:**

1. Empirical Benchmarks Missing Key Baselines: While the paper compares with strong RL baselines, it does not provide direct quantitative comparisons to major previous length-efficiency methods: Length-Controlled Policy Optimization (LCPO) [1], Token-budget [2] , Park et al [3]
[1] P. Aggarwal and S. Welleck, “L1: Controlling how long a reasoning model thinks with reinforcement learning,” in Second Conference on Language Modeling, 2025.
[2] T. Han, Z. Wang, C. Fang, S. Zhao, S. Ma, and Z. Chen, “Token-budget-aware LLM reasoning,” in Findings of the Association for Computational Linguistics: ACL 2025
[3] R. Park, R. Rafailov, S. Ermon, and C. Finn, “Disentangling length from quality in direct preference optimization,” in Findings of the Association for Computational Linguistics: ACL 2024
2. While tables and longitudinal curves cover length/accuracy tradeoffs, there is little analysis of response quality from a human perspective. Are the concise responses still interpretable, pedagogically sound, or merely “shorter”? Occasional “over-compression” could degrade user interpretability.
3. Missing Related Work: Several recent works highly relevant to chain-of-thought efficiency [1], adaptive reasoning [3,4] are not cited or positioned against, limiting contextualization and novelty assessment.
[1] S. Goyal, Z. Ji, A. S. Rawat, A. K. Menon, S. Kumar, and V. Nagarajan, “Think before you speak: Training language models with pause tokens,”
[2] X. Chen, S. Zhou, K. Liang, and X. Liu, “Distilling reasoning ability from large language models with adaptive thinking,” IEEE Transactions on Neural Networks and Learning Systems, pp. 1–14, 2025.
[3] G. Liang, L. Zhong, Z. Yang, and X. Quan, “Thinkswitcher: When to think hard, when to think fast,

**Questions:**

See weaknesses

---

### Official Review · Reviewer_EqgA · 2025-10-31

**Soundness:** 4
**Presentation:** 4
**Contribution:** 3
**Rating:** 4
**Confidence:** 4

**Summary:**

This paper proposes an RL-based method to prevent large language models from performing excessive reasoning during chain-of-thought (CoT) generation. The proposed framework unifies verifiable and fuzzy tasks under a single pairwise reward formulation, supported by clear theoretical assumptions. The authors demonstrate that the proposed approach can maintain accuracy while significantly reducing reasoning length across various datasets and model scales. Notably, the method can be seamlessly integrated with existing RL frameworks such as GRPO without modification.

**Strengths:**

1. $\textbf{Simple and intuitive formulation:}$
The proposed method can be applied without modifying the reinforcement learning framework or training a separate reward model. Its design based on pairwise reward comparison makes it easily extendable to different reward structures.

2. $\textbf{Effective control of reasoning length:}$
Across multiple experiments, the method consistently achieves shorter reasoning chains than standard RL or SFT baselines while maintaining comparable accuracy. This shows its capability to suppress unnecessary reasoning during inference.

3. $\textbf{Versatility across reward types:}$
The framework effectively handles both verifiable and fuzzy tasks by decoupling task rewards and length rewards, while still considering their interactions. This generalization potential is empirically supported by experiments on both mathematical reasoning and instruction-following tasks.

**Weaknesses:**

1. $\textbf{Limited comparison with length reduction methods:}$
While Appendix B.3 includes partial comparisons with several length-reduction methods such as Long2Short (Team et al., 2025), ThinkPrune (Hou et al., 2025), and DAST (Shen et al., 2025), the evaluation scope remains quite limited.
Recent studies such as Length Controlled Policy Optimization (LCPO) (Aggarwal & Welleck, 2025), O1-Pruner (Luo et al., 2025a), and Training Language Models to Reason Efficiently (Arora & Zanette, 2025) have proposed alternative approaches to reasoning-length control.
Importantly, the present work shares several appealing properties with these methods—notably, framework simplicity, compatibility with existing RL algorithms, and minimal reliance on additional reward models.
Highlighting these common strengths while expanding empirical comparisons would help position this paper more clearly within the broader landscape of efficient reasoning research.
A more comprehensive quantitative comparison with these latest approaches would also strengthen the empirical validity of the work and clarify the trade-off between reasoning-length reduction and performance retention.


2. $\textbf{Weak analysis of training dynamics:}$
Although the experiments vary datasets and models extensively, there is limited analysis of intermediate training behavior — for example, how the number of correct responses evolves or how the reward distribution stabilizes. Such analysis could help illuminate how the proposed pairwise reward formulation stabilizes RL training and reduces CoT length.

**Questions:**

1. How does the performance vary with different hyperparameter settings (α, β)? Appendix B.2 provides limited results, but a more systematic analysis of stability and sensitivity would be helpful.
2. How does the proposed approach compare quantitatively to other recent length-reduction algorithms (e.g., LCPO, O1-Pruner) on common benchmarks?
3. In Figure 7, the baseline also shows decreasing reasoning length during training, which contrasts with the usual trend of length expansion in RL. Could the authors clarify the cause (e.g., reward saturation or sampling bias)?
4. What happens if α = 0 and β > 0? Would the model still show a preference for shorter correct responses indirectly through β?
5. How does accuracy on verifiable tasks change throughout training compared to baselines? This could reveal whether the method mainly affects early-stage or late-stage learning behavior.

---

### Official Review · Reviewer_39Vc · 2025-11-01

**Soundness:** 4
**Presentation:** 4
**Contribution:** 4
**Rating:** 4
**Confidence:** 4

**Summary:**

This paper introduces a novel reward framework to address the inefficiency of CoT reasoning in large language models, where models often "overthink" simple problems. Instead of directly penalizing response length, the authors propose a method based on pairwise comparisons between generated responses. This approach constructs rewards by evaluating both the quality (correctness) and conciseness (length) of responses relative to others in a sampled batch. The method is grounded in a clear set of theoretical assumptions and is elegantly extended from verifiable tasks with ground truth to "fuzzy" tasks requiring subjective evaluation. Extensive experiments on various reasoning benchmarks with different model sizes demonstrate that the proposed algorithm significantly reduces CoT length while maintaining or even slightly improving task accuracy.

**Strengths:**

1. The paper tackles a highly practical and underexplored issue in RL-based reasoning: the inefficiency of CoT overthinking, which is both timely and relevant.

2. The proposed pairwise reward formulation is simple, elegant, and model-agnostic, avoiding the need for additional critics or external supervision.

3. Results convincingly show that the method significantly shortens reasoning chains while maintaining or even improving accuracy.

**Weaknesses:**

1. The main innovation is relatively small, focusing on a new reward formulation.

2. The framework’s performance appears sensitive to α and β, and the paper lacks clear discussion on how to choose them or how robust the method is to their variation.

3. The pairwise comparison step scales quadratically with the group size N, which may limit efficiency for large datasets, yet this trade-off is not discussed.

4. The paper’s structure and writing are somewhat verbose, which makes the core idea harder to grasp.

5. Missing baseline comparisons, such as cosine reward [1].

6. In Figure 3, the final data point for the "1.5b(baseline)" at 1000 steps is missing across all subplots. This omission prevents a complete comparison at the end of training and should be corrected for clarity.

[1] Demystifying Long Chain-of-Thought Reasoning in LLMs

**Questions:**

See weakness.

---

### Meta-Review · Area_Chair_Ehmp · 2026-01-02

**Summary:**

This paper proposes a RL framework to reduce unnecessary CoT verbosity in LLMs. Rather than explicitly penalizing response length, the method defines rewards through pairwise comparisons between sampled responses, prioritizing correctness and conciseness among correct answers. The framework is theoretically motivated and applies to both verifiable tasks with ground truth and fuzzy tasks relying on subjective comparisons. The results consistently show substantial reductions in reasoning length while largely maintaining accuracy, which reviewers viewed as a meaningful practical outcome. At the same time, reviewers raised several concerns. The core contribution is relatively incremental, centered on a new reward formulation rather than a fundamentally new learning paradigm. Multiple reviewers noted missing or limited comparisons with closely related recent work on reasoning-length control (e.g., LCPO, O1-Pruner, Long2Short-style methods), which makes it difficult to clearly assess empirical advantages. Reviewers also pointed to insufficient analysis of training dynamics and hyperparameter sensitivity, particularly regarding the stability and robustness of the method. For fuzzy tasks, reviewers questioned whether aggressively shortened or near-zero CoT outputs preserve interpretability and whether the evaluation fully demonstrates adaptive reasoning rather than over-compression. Finally, several reviewers mentioned presentation issues, including verbosity and organization, that obscure the core ideas.

**Reviewer Concerns:**

There was no author rebuttal, so no reviewer concerns were addressed.

**Reviewer Scores:**

Based on the lack of rebuttal, it is unlikely that reviewer scores would have changed.

---

### Decision · Program_Chairs · 2026-01-26

Reject